# Corrective Unlearning for MRI Reconstruction

**Abstract.** Magnetic Resonance Imaging reconstruction accelerates image acquisition by reconstructing high-quality images from undersampled k-space data using deep learning. However, real-world deployment of these models remains hindered by concerns around trustworthiness, generalization, and data privacy, especially in the presence of corrupted or adversarial training samples. We propose a Corrective Machine Unlearning framework that selectively removes the influence of harmful data while preserving overall model performance. By leveraging techniques such as Selective Synaptic Dampening, our approach aims to robustly and efficiently forget poisoned representations. Experimental results on MRI reconstruction tasks demonstrate that Corrective Machine Unlearning can effectively mitigate artifacts introduced through data poisoning while maintaining high fidelity on untainted inputs. Our findings underscore the promise of corrective unlearning as a practical step toward safer, privacy preserving, and clinically reliable MRI systems. All code and scripts used are available at Github repository.

**Keywords:** Medical Imaging · Corrective Machine Unlearning · MRI Reconstruction · Deep Learning · Selective Synaptic Dampening · Trustworthy AI · MRI Acceleration

## 1 Introduction

Magnetic Resonance Imaging (MRI) is essential in clinical diagnostics but is hindered by slow acquisition speeds. Deep learning–based accelerated reconstruction techniques, such as variational networks[16], have significantly improved image quality from undersampled data.

However, these models remain vulnerable to corrupted training samples and out-of-distribution inputs[22], which can lead to hallucinated anatomical structures and unreliable reconstructions. These risks are further intensified when using large, noisy, or biased datasets. Moreover, regulatory frameworks like GDPR demand mechanisms to remove specific data influence, motivating the need for responsible model correction.

We propose *Corrective Machine Unlearning* as a post-training solution to selectively remove the influence of flawed data without full retraining. Using methods like Selective Synaptic Dampening (SSD), we target improved robustness and clinical viability in MRI reconstruction models.

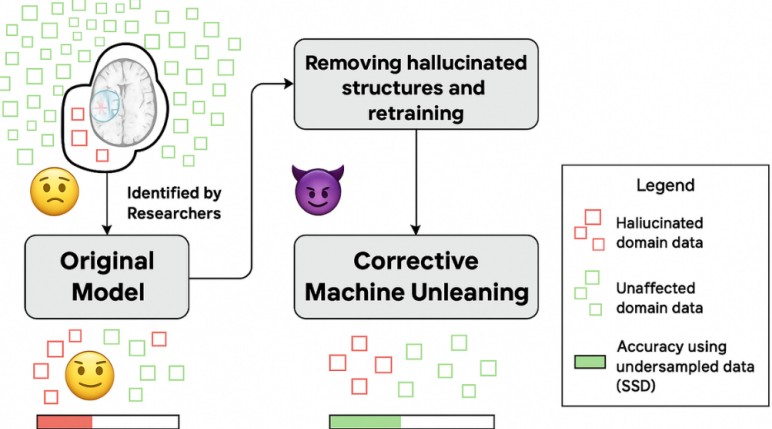

Fig. 1: Conceptual overview of corrective unlearning to mitigate hallucinations from corrupted training data.

## 2    Related Work

*Litjens et al.* [11] and *Shen et al.* [15] demonstrated the transformative impact of deep learning in medical imaging, particularly for tasks such as segmentation, detection, and disease classification in various modalities. Among these, MRI reconstruction has seen significant advancement, moving beyond classical compressed sensing [12] to end-to-end deep learning frameworks such as variational networks proposed by *Hammernik et al.* [9], which combine data fidelity with learned priors to achieve accelerated and high-quality reconstructions.

However, such models often lack robustness to out-of-distribution or corrupted inputs, leading to hallucinated features and clinical risks. *Xu et al.* [20] and *Wang et al.* [19] proposed machine unlearning to remove specific data influence without full retraining, aligning with privacy mandates like GDPR [18].

Based on this, *Goel et al.* [8] introduced corrective unlearning, which deals with scenarios involving only partial knowledge of corrupted or adversarial data. Their strategies, such as selective parameter damping, offer promising solutions to mitigate the lasting impact of such data on model behavior. Existing approaches [21] have considered the problem of privacy based unlearning. Here we extend this approach to corrective unlearning bridging the gap to real world applications where both clinical accuracy and regulatory compliance are important.

# 3 Methods

## 3.1 Problem Formulation

Following [8], let $D$ be the full training dataset used to train a model $M$ with parameters $\theta$. We define a *forget set* $D_f \subset D$, of which a subset $D_f^* \subset D_f$ is available for unlearning. The *retain set* is $D_r = D \setminus D_f$.

Our objective is to obtain an updated model $M'$ with parameters $\theta'$, such that the influence of $D_f$ is removed while maintaining performance on $D_r$.

1. Performance Retention on $D_r$
2. Performance Degradation on $D_f$
3. Model Efficiency

To represent this trade-off, we define a score function:

$$\mathcal{S}(M, \theta, D_f; \alpha) = \alpha \cdot \mathrm{Perf}(M, \theta; D_r) + (1 - \alpha) \cdot [1 - \mathrm{Perf}(M, \theta; D_f)],$$

where $\mathrm{Perf}(\cdot)$ is a performance metric (e.g., accuracy or loss), and $\alpha \in [0, 1]$ controls the relative importance of retention and forgetting. The optimization objective is then:

$$\theta^* = \arg\max_{\theta} \ \mathcal{S}(M, \theta, D_f^*; \alpha), \ \lambda^* = \arg\max_{\lambda \in R^n} \ \mathcal{S}(M(\lambda), \theta, D_f; \alpha)$$

where $\lambda$ represents model-specific hyperparameters.

## 3.2 Unlearning Approaches

We consider several corrective machine unlearning strategies suited for the medical imaging domain, particularly for MRI reconstruction tasks:

1. **Retraining from Scratch:** The model is retrained solely on the retain set:

   $$\theta' = \arg\min_{\theta} \mathcal{L}(\theta; D_r).$$

   While this method fully removes $D_f$, it is computationally expensive and impractical for clinical use. Moreover, limited availability of forget data can significantly degrade model performance.
2. **Selective Synaptic Dampening (SSD):** This approach[7] uses the Fisher Information Matrix to identify and dampen parameters most influenced by the forget set:

   $$F_{D_f} = E_{x \sim D_f} \left[ \nabla_\theta \log p(x|\theta) \nabla_\theta \log p(x|\theta)^\top \right].$$
3. **Bad Teacher Distillation:** Following [6], a biased teacher model overfit to $D_f$, is first trained. The student model then subtracts this overlearned signal to retain only the unbiased representation.
4. **Gradient Ascent (GA):** A targeted gradient ascent step[17] is performed on the loss over $D_f$, with learning rate $\eta$, to negate its influence:

   $$\theta' = \theta + \eta \nabla_\theta \mathcal{L}(\theta; D_f).$$

   Careful tuning of $\eta$ is critical to reverse learning without destabilizing the model.

## 4   Dataset

**M4Raw:** M4Raw[13] is a 28 GB, multi-anatomy MRI dataset providing raw multi-coil k-space data, magnitude images, sensitivity maps, and undersampling masks. It covers regions such as the brain and knee, all acquired with fastMRI-style protocols, and is released under an open license to support reproducible acceleration experiments across a variety of clinical scenarios.

**EXBox1:** EXBox1[5] is a publicly released collection of artifact-heavy MR scans exhibiting ghosting, motion blur, and intensity distortions. Each scan comes with paired clean and corrupted versions, enabling precise evaluation of unlearning methods under real-world acquisition anomalies without sacrificing anatomical diversity.

**BraTS:** The BraTS dataset[4] provides T1-weighted MRI scans of glioma tumors spanning multiple grades. Though originally released as spatial-domain images, we convert each volume to synthetic k-space via Fourier transform and inject adversarial perturbations and label noise to create poisoned examples for robust unlearning assessment.

### Poisoning Techniques

We first explored a mislabeling-based attack to mimic potential misannotations in loosely curated datasets. Initial attempts using basic PGD attacks to simulate tumor-like features in healthy scans resulted in noisy perturbations. To improve realism, we developed a multi-objective adversarial approach combining SSIM loss, total variation loss, and region-aware masking. However, for our experiments, we focused solely on mislabeling attacks.

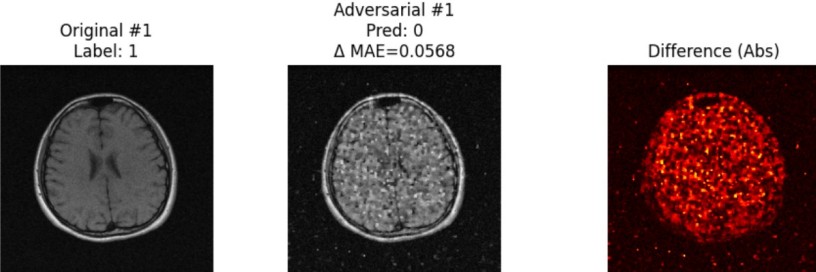

Fig. 2: Adversarial attack on a sample: Original image (Label: 1) on the left and its adversarial counterpart (Predicted: 0) on the right. Minor structural noise led to class flipping.

## 5   Experiments

### 5.1   Classifier Experiment

To assess whether unlearning was necessary, we first tested if a significant portion of the forget set could be identified outright. If so, retraining from scratch—after

removing these samples—could suffice without a dedicated unlearning strategy. Even with a powerful model like ResNet-50[10] pretrained on 1M images, accu-

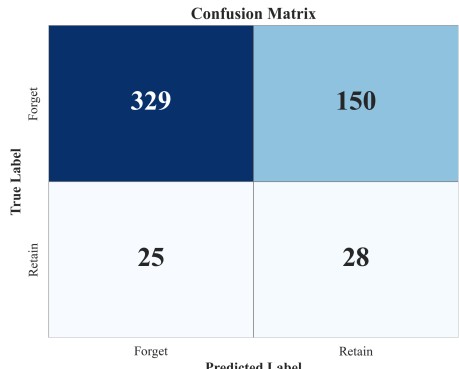

Fig. 3: Confusion matrix for the Forget Classifier (RNet). True label vs. predicted label. The model performs well on 'retain' samples but struggles to correctly classify many 'forget' instances.

rate detection of anomalous or poisoned samples remains challenging. As shown, a significant number of 'forget' instances are misclassified, meaning retraining on a filtered set would still leave many corrupted samples intact.

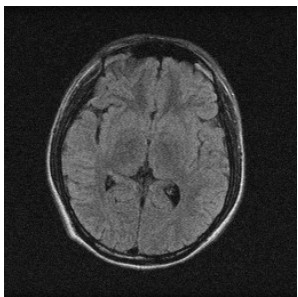
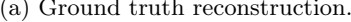
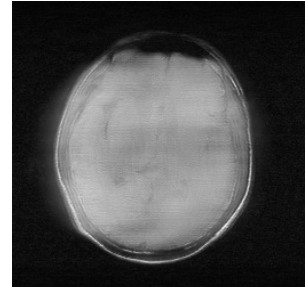

(a) Ground truth reconstruction.  (b) Poisoned model reconstruction.

Fig. 4: Visual comparison of clean and poisoned MRI reconstructions.

## 5.2  Unlearning Experiments

To evaluate the corrective unlearning strategies, we compared five methods using a known subset of poisoned samples as the forget set, and the remaining clean data as the retain set. An oracle model, trained from scratch on the unpoisoned dataset, served as the reference baseline.

Finetuning methods were run to convergence on the retain set, starting from the poisoned checkpoint. In contrast, SSD was applied for a single epoch to highlight its immediate corrective effect.

**Experimental Setup**

1. **Dataset partitioning:** For each target poison level (e.g., 1%, 5%, 10%), we randomly sampled that fraction of the training set, and poisoned it to form the true forget set. From this, we designated a known forget subset (50% of the true forget set) available to each unlearning method; the remainder of the data constituted the retain set.
2. **Baseline (no unlearning):** The original poisoned model, trained on the full dataset including all poisoned samples.
3. **Oracle:** A clean model trained on entirely unpoisoned data, serving as the best-achievable performance.
4. **Evaluation:** We measured reconstruction quality (e.g. PSNR, SSIM) separately on the true retain set(unpoisoned samples)mcomplete forget set and the validation set.

# 6 Results and Interpretation

## 6.1 Unlearning Experiment Results

We conducted a series of unlearning experiments to evaluate the effectiveness of various methods in mitigating the impact of poisoned data on a reconstruction task. The experiments focused on a range of poisoned model percentages, with metrics evaluated on both the forget and retain sets. Below, we report the results of our most promising approach, Selective Synaptic Dampening (SSD), alongside a comparison with the original model's unlearning performance.

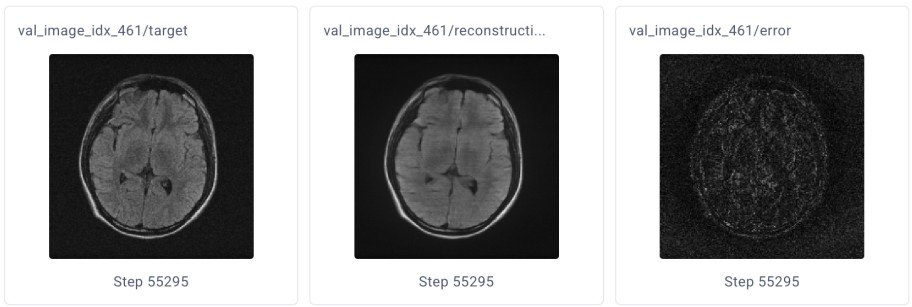

Fig. 5: Reconstructions obtained after retrain from scratch

**Metrics on Forget and Retain Sets** We evaluated all methods on 9,216 samples using a model poisoned with 30% corrupted data, where 20% of the forget set was known a priori. Inference ran at 12.19 iterations per second. As

shown in Figure 6, the original model achieves the best reconstruction performance across SSIM, PSNR, and NMSE, reflecting high fidelity even on poisoned data. In contrast, our SSD method achieves a favorable trade-off, significantly degrading performance on the forget set while maintaining high accuracy on the retain set.

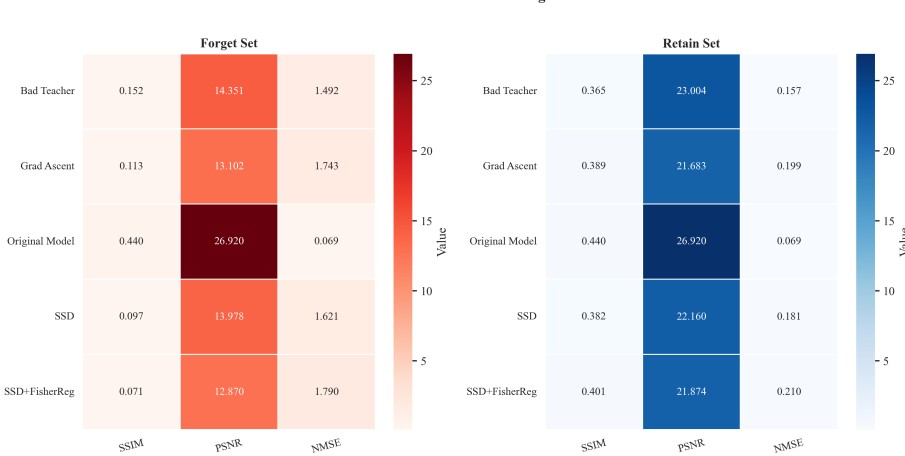

Fig. 6: Heatmap comparison of SSIM, PSNR, and NMSE across forget and retain sets for different unlearning methods. SSD demonstrates strong forgetting while preserving retain performance.

**Selective Synaptic Dampening (SSD)** SSD achieved this performance using a single hyperparameter configuration. Its mechanism selectively suppresses synaptic weights to forget poisoned information while preserving useful features. Given its sensitivity to $\alpha$ and $\lambda$, we anticipate further gains through systematic hyperparameter tuning.

### 6.2   Retention of Generalizability in the Unlearnt Model

A key finding in our experiments is the ability of the unlearnt model to revert to clean reconstructions, even when trained with poisoned data. As shown in Figure 7, the output of the model post-unlearning closely resembles the ground-truth image rather than the corrupted input it was originally exposed to. This suggests that our Selective Synaptic Dampening (SSD) approach effectively removes the influence of harmful training signals while preserving useful representational capacity.

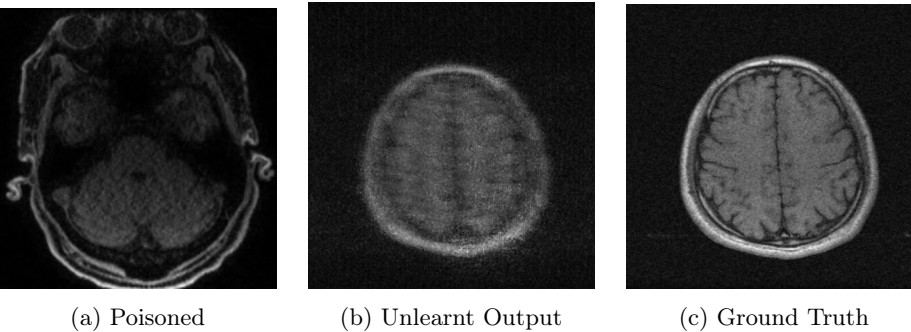

(a) Poisoned                (b) Unlearnt Output                (c) Ground Truth

Fig. 7: Comparison of poisoned input, unlearnt model output, and clean ground-truth reconstruction.

This behavior demonstrates SSD's robustness to adversarial perturbations, its ability to retain generalizable features, and its precision in selectively forgetting only the corrupted influences key attributes for reliable deployment in clinical settings.

## 7    Conclusion

We presented a corrective machine learning framework for deep learning-based MRI reconstruction, which allows the selective removal of corrupted training data while preserving performance on clean inputs. Our approach, centered on selective synaptic dampening (SSD), and supported by complementary techniques like gradient ascent and bad teacher distillation, proved effective against various data corruptions, including poisoning and adversarial attacks.
SSD consistently mitigated artifacts and restored reconstruction quality, demonstrating the utility of unlearning beyond privacy - improving model trust and robustness in clinical settings. As regulatory requirements like GDPR grow in importance, such scalable unlearning methods offer a practical pathway for safe and reliable medical AI.
Future directions include adaptive dampening strategies, improving reconstruction fidelity, integration with federated learning, and expert-guided clinical validation.

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
