# OpenReview forum: "Corrective Unlearning for MRI Reconstruction"
_MICCAI.org/2025/Workshop/MSB_EMERGE — Submitted to MSB EMERGE 2025_

### Official Review · Reviewer_CcA5 · 2025-07-02

**Recommendation:** 1
**Confidence:** 4

**Clarity:**

The paper is unclear and difficult to understand due to significant clarity issues, major revision is necessary

**Feedback:**

Data consistency is particularly relevant in reconstruction tasks, as it helps avoid hallucinations and ensures reliable outputs. The authors should explicitly relate their method to this aspect.

Please introduce and define abbreviations such as GDPR when first mentioned.

The score function uses "1 - perf" (performance metric, e.g. accuracy or loss.). what is the value rang of the performance metric?

In Figure 2, clarify what the classes represent.

Explain the unlearning experiments more thoroughly and also clearly describe all poisoning strategies employed.

List all evaluation metrics used in the study explicitly.

Be cautious with wording, e.g., referring to the method as "our SDD approach" may either compromise anonymity or is incorrect since the authors did not propose the SDD.

The conclusion needs significant revision. For instance, the claim that the method mitigated artifacts and restored reconstruction quality is not supported by Figure 7.

The authors use the term “privacy” but did not discuss how the approach aligns with privacy objectives and the term remains vague.

Please consider providing an anonymous GitHub repository to facilitate reproducibility and further review.

Is it right that the detection of poisoned samples remains challenging and the poisoned data (or at least 50%) needs to explicitly be defined. Please clarify and discuss the implications of this assumption.

**Justification:**

I recommend rejection of this paper due to several substantial shortcomings in both conceptual clarity and execution. I do not believe the paper meets the standards for publication at this time. Substantial revision is required to clarify the problem setting and proposed method, improve experimental rigor, and validate the method's contributions.

**Reproducibility:**

Not enough amount of details available for reproducing the main results, and open access details are unclear

**Strengths:**

While the paper addresses the important problem of mitigating harmful data influence through corrective unlearning, the current paper, proposed approach and experimental results indicate substantial room for improvement.

**Summary:**

The authors propose a corrective unlearning framework designed to remove the influence of harmful data without necessitating retraining the entire network. The primary objective is to maintain the model's overall performance while degrading its effectiveness on poisoned data. However, their experimental results show that the original model (without any unlearning applied) still achieved superior performance across all three metrics, even when poisoned data was included.

**Weaknesses:**

**Mismatch with Stated Goals**: The paper claims to be a reconstruction approach, yet it does not include any explicit reconstruction tasks or evaluations. While reading I was not sure if its meant to maybe address denoising?. Overall the “reconstruction” problem and application area is not defined at all.

**Insufficient Related Work**: The introduction and related work sections are incomplete, making it challenging to accurately position the paper within existing literature. This omission limits the reader's ability to understand the relevance and significance of the proposed approach.

**Lack of Novelty and Clarity in Methodology**: The proposed method's novelty is unclear, and the paper does not adequately distinguish its contributions from existing approaches like
[8] Corrective Machine Unlearning
[21] Erase to Enhance: Data-Efficient Machine Unlearning in MRI Reconstruction.

**Ambiguity Regarding Data**: The paper fails to clearly describe the final data that is used, complicating replication or assessment of the presented results. It is not clear how to the different datasets are used. Also modifications of the data (e.g. making samples poisoned) is not explained at all.

**Experimental Setup Unclear**: The experimental design and setup suffer from significant clarity issues. Crucially, details regarding the neural network architecture, training procedures, and overall experimental pipeline are inadequately documented, severely impacting reproducibility. Additionally, essential experimental conditions and settings remain unspecified, further obscuring the interpretation of the results.

**Poor Experimental Results**: Results indicate that the baseline method consistently outperforms the proposed approach across all evaluated metrics, casting doubt on the utility of the proposed method.

---

### Official Review · Reviewer_JVpx · 2025-07-02

**Recommendation:** 1
**Confidence:** 4

**Clarity:**

The paper has significant clarity issues that hinder understanding, substantial revision is required to improve clarity

**Feedback:**

The problem is highly relevant, but the paper suffers from unclear task definition, unclear novelty beyond prior work, missing experimental details, contradictory results, and no real privacy validation. As it stands, it is too incomplete and the main claims are not supported by the experiments. With significant revision and deeper evaluation, the idea could still be worth pursuing.

**Justification:**

Based on the points mentioned above, I suggest that the paper is not ready for publication in the current format.

**Reproducibility:**

Not enough amount of details available for reproducing the main results, and open access details are unclear

**Strengths:**

**Addresses a real problem**: The threat of harmful training data is serious in clinical MRI AI. Hallucinated anatomy due to poisoning is a genuine risk.
**Idea is reasonable**: Trying to “subtract out” poisoned information without full retraining makes sense if done properly.*
**Good datasets**: Using M4Raw and EXBox1 adds practical realism. Including BraTS for simulated tumors is interesting.

**Summary:**

The paper proposes a “Corrective Machine Unlearning” method for MRI reconstruction. The goal is to selectively forget harmful or poisoned training data without retraining the full model. The authors highlight real challenges in MRI AI such as mislabels, adversarial poisoning, and regulatory “right to be forgotten” requirements (GDPR). The proposed approach centers on Selective Synaptic Dampening (SSD), with extra variants like Bad Teacher Distillation and Gradient Ascent. They test on three MRI datasets using synthetic poisoning and compare to baselines.

**Weaknesses:**

I’ll break this by sections for clarity:
1. Title & Abstract
1.1. The title says MRI Reconstruction but the core task isn’t defined rigorously.
1.2. The abstract mentions “high-fidelity images” but the actual results show fidelity drops after unlearning.
1.3. The code link is given, but the paper does not prove whether the code fully reproduces the pipeline.

2. Related Work
2.1. Missing deeper discussion of domain-specific unlearning papers; e.g. Erase to Enhance is only mentioned once.
2.2. No clear statement on how SSD improves over Goel et al. (2024) for this domain.
2.3. Related medical imaging defenses (like robust training or anomaly detection) are skipped.

3. Methods and Results
3.1. The central task is ill-defined and confusing. The paper claims to target MRI reconstruction, but never clearly specifies:
Are reconstructions performed directly in k-space? In image domain? How is data consistency enforced?
No mention of standard reconstruction baselines or benchmarks (e.g., vs. CS-MRI, vs. variational networks, vs. modern deep MRI models).
3.2. The paper sometimes drifts into treating the task like classification (e.g., with confusion matrices in Figure 3) and sometimes like reconstruction, which adds to the confusion.

4.  The method is borrowed almost directly, but the novelty claim is vague
4.1. The main method (SSD) is from prior work, and “corrective machine unlearning” is from Goel et al. (2024).
4.2. The paper doesn’t clarify what exactly is new: Is it adapting SSD to MRI? That’s not discussed in detail; Is it the way they define the forget and retain sets? The paper says 50% of the forget set is known a priori but doesn’t justify why or how.

5.  Weak results that contradict the main claim
5.1. Figure 6 shows that the poisoned baseline model actually outperforms the unlearnt models on standard metrics (SSIM, PSNR, NMSE).
5.2. The “oracle” model (trained on fully clean data) obviously does best, but SSD fails to come close.
5.3. Figure 7 shows a single qualitative result that is claimed to look better, but no radiologist or human evaluation is provided.

---

### Official Review · Reviewer_uo3t · 2025-07-08

**Recommendation:** 1
**Confidence:** 4

**Clarity:**

The paper is unclear and difficult to understand due to significant clarity issues, major revision is necessary

**Feedback:**

Please refer to my comments on the weaknesses, where I provide specific suggestions to improve the clarity, robustness, and self-containedness of the work.

**Justification:**

While the paper addresses an important topic, it falls short in several critical areas that undermine its overall contribution and clarity, such as unclear contributions and novelty, insufficient experimental justification, unsupported claims, and a lack of discussion on limitations. Therefore, I believe the paper is not ready for publication and requires major revisions.

**Reproducibility:**

Not enough amount of details available for reproducing the main results, and open access details are unclear

**Strengths:**

This paper addresses a relevant problem: MRI reconstruction models are prone to hallucinations caused by mislabeled, harmful, or poisoned data, as they often lack robustness to out-of-distribution or corrupted inputs. Therefore, studying how to reduce the impact of such data is important for such tasks.

**Summary:**

This paper introduces a Corrective Machine Unlearning method to remove the influence of mislabeled, harmful or poisoned data in MRI reconstruction models—without requiring retraining. The authors explore the following unlearning strategies applicable to MRI reconstruction: Selective Synaptic Dampening (SSD), Bad Teacher Distillation, and Gradient Ascent (GA). They evaluate the effectiveness of these strategies by measuring the performance of the unlearning-applied model on both the forget set (poisoned data to be unlearned) and the retain set (data to be preserved), using SSIM, PSNR, and NMSE as evaluation metrics.  These results are then compared to those of a baseline model trained without unlearning on the unpoisoned dataset. The baseline model outperforms all unlearning-applied models across metrics.  However, the authors highlight that the SSD-applied model achieves strong performance in terms of the PSNR on the retain set (SSD PSNR 22.160 vs. baseline model PSNR= 26.920) while showing low fidelity on the forget set (SSD PSNR 13.978 vs. baseline model PSNR= 26.920) (see Figure 6).

**Weaknesses:**

-  The specific contributions and novelty of this paper remain unclear. For example, at the end of the related work section, the authors briefly mention two prior works ([8],[21]) they build upon but do not clearly articulate how their approach differs from or advances beyond them. More critically, the introduction states: *"We propose Corrective Machine Unlearning as a post-training solution to selectively remove the influence of flawed data without full retraining."* This statement is potentially misleading, as it does not acknowledge or cite the original work that introduced this concept. This omission is particularly problematic given that Figure 1 appears to be an adapted version of Figure 1 from the original work [8], yet no attribution is provided. A more thorough discussion in the related work section is necessary, including a precise comparison with the referenced studies and provide attribution for adapted figure.

-  The interpretation of the results lacks depth. The authors appear to rely on accuracy-based verification to evaluate the effectiveness of unlearning; however, this is neither explicitly stated nor complemented by alternative verification methods. Since assessing the success of the unlearning process is a non-trivial task, clarifying the evaluation approach is essential—not only for understanding the impact of the results but also for ensuring the self-containedness of the manuscript.

-  The authors suggest that the SSD-applied model successfully suppresses the effects of harmful training signals while maintaining useful representational capacity. However, the evidence supporting this claim is limited to a single example (Figure 7), which lacks justification or explanation regarding how the figure substantiates the assertion. This is insufficient to support such a strong claim. Additional empirical evidence and a clearer rationale are necessary to convincingly validate this conclusion.

-  The manuscript lacks a dedicated section discussing the limitations of the proposed approach. For instance, it remains unclear whether any of the unlearning strategies discussed can be applied in scenarios where practitioners do not have (even partial) access to the corrupted input data. This needs to be addressed.

-  The authors mention privacy and GDPR as motivating factors for their work, but this connection is not clearly reflected in their experiments or their interpretation. A precise definition of what privacy means in this context, along with a clear explanation of the privacy guarantees offered by their approach, is necessary.